# Machine learning outperformed logistic regression classification even with limit sample size: A model to predict pediatric HIV mortality and clinical progression to AIDS

**Sara Domínguez-Rodríguez**[1], **Miquel Serna-Pascual**[1], **Andrea Oletto**[2],
**Shaun Barnabas**[3], **Peter Zuidewind**[3], **Els Dobbels**[3], **Siva Danaviah**[4], **Osee Behuhuma**[4],
**Maria Grazia Lain**[5], **Paula Vaz**[5], **Sheila Fernández-Luis**[6,7], **Tacilta Nhampossa**[7],
**Elisa Lopez-Varela**[7], **Kennedy Otwombe**[8], **Afaaf Liberty**[8], **Avy Violari**[8], **Almoustapha
Issiaka Maiga**[9], **Paolo Rossi**[10], **Carlo Giaquinto**[11], **Louise Kuhn**[12], **Pablo Rojo**[1],
**Alfredo Tagarro**[1,13,14]*, **on behalf of EPIICAL Consortium**¶

1 Pediatric Infectious Diseases Unit, Fundación para la Investigación Biomédica del Hospital 12 de Octubre, Madrid, Spain, 2 PENTA Foundation, Padova, Italy, 3 Family Centre For Research With Ubuntu (FAMCRU), Stellenbosch University, Cape Town, South Africa, 4 Africa Health Research Institute (AHRI), Durban, South Africa, 5 Fundação Ariel Glaser contra o SIDA Pediátrico, Maputo, Mozambique, 6 Centro de Investigação em Saúde de Manhiça (CISM), Maputo, Mozambique, 7 Barcelona Institute for Global Health (ISGLOBAL), Barcelona, Spain, 8 Perinatal HIV Research Unit, Faculty of Health Sciences, University of the Witwatersrand, Johannesburg, South Africa, 9 Gabriel Toure University Hospital, Bamako, Mali, 10 Division of Immune and Infectious Diseases, Istituto di Ricovero e Cura a Carattere Scientifico, Ospedale Pediatrico Bambino Gesu, Rome, Italy, 11 Department of Surgery, Oncology and Gastroenterology, Section of Oncology and Immunology, University of Padova, Padova, Italy, 12 Gertrude H. Sergievsky Center, Vagelos College of Physititlicians and Surgeons, Columbia University Irving Medical Center, New York, NY, United States of America, 13 Universidad Europea de Madrid, Madrid, Spain, 14 Fundación para la Investigación e Innovación Biomédica del Hospital Universitario Infanta Sofía, Hospital Universitario Infanta Sofía, San Sebastián de los Reyes, Madrid, Spain

¶ Membership of the EPIICAL Consortium is provided in the Acknowledgments.
* alfredotagarro@gmail.com

## Abstract

Logistic regression (LR) is the most common prediction model in medicine. In recent years, supervised machine learning (ML) methods have gained popularity. However, there are many concerns about ML utility for small sample sizes. In this study, we aim to compare the performance of 7 algorithms in the prediction of 1-year mortality and clinical progression to AIDS in a small cohort of infants living with HIV from South Africa and Mozambique. The data set (n = 100) was randomly split into 70% training and 30% validation set. Seven algorithms (LR, Random Forest (RF), Support Vector Machine (SVM), K-Nearest Neighbor (KNN), Naïve Bayes (NB), Artificial Neural Network (ANN), and Elastic Net) were compared. The variables included as predictors were the same across the models including sociodemographic, virologic, immunologic, and maternal status features. For each of the models, a parameter tuning was performed to select the best-performing hyperparameters using 5 times repeated 10-fold cross-validation. A confusion-matrix was built to assess their accuracy, sensitivity, and specificity. RF ranked as the best algorithm in terms of accuracy (82,8%), sensitivity (78%), and AUC (0,73). Regarding specificity and sensitivity, RF showed better performance than the other algorithms in the external validation and the

**Data Availability Statement:** Data cannot be shared publicly because of General Data Protection Regulation (EU GDPR). Data are available from the REDCap Institutional Data Access / Ethics Committee (contact carla.paganin@opbg.net, PENTA-ID Foundation (http://penta-id.org/)) for researchers who meet the criteria for access to confidential data.

**Funding:** This work has been supported within EPIICAL project through an independent ViiV grant to the PENTA (Paediatric European Network for Treatment of AIDS) Foundation. The funders had no role in study design, data collection, analysis, interpretation, or manuscript preparation.

**Competing interests:** The authors have declared that no competing interests exist.

highest AUC. LR showed lower performance compared with RF, SVM, or KNN. The outcome of children living with perinatally acquired HIV can be predicted with considerable accuracy using ML algorithms. Better models would benefit less specialized staff in limited resources countries to improve prompt referral in case of high-risk clinical progression.

## Introduction

There is still a high rate of mortality in children living with HIV in the first years after antiretroviral (ART) initiation in Sub-Saharan Africa. These rates reported being as high as 21% in South Africa and 26% in Mozambique [1].

Several risk factors are associated with increased childhood mortality and clinical progression to AIDS. Advanced disease at HIV diagnosis [2], older age at ART initiation [3], higher baseline HIV viral load [4], low weight-for-age Z-score [3], low % CD4 [4], and female sex [3] are the factors reported most. However, the risk models that yielded these risk factors were built using logistic regression (LR) models, considered the standard and most used approach for binary classification in the context of low-dimensional data.

Modern ML algorithms use statistical, probabilistic and optimization methods to detect patterns or rules from large and complex datasets [5]. Specifically, supervised ML systems create prediction models by studying a dataset where the outcome is known and subsequently validating this model in a dataset where the outcome is unknown. Unlike classical multivariable LR, ML allows the inclusion of a large number of predictors, that may exceed the number of observations, and allows the combination of these predictors in flexible linear and nonlinear ways [6]. By relaxing model assumptions, ML may improve risk classification.

Despite its advantages, the application of ML other than LR in the HIV field has been limited [7–11]. Several studies have claimed the better performance of machine learning versus logistic regression in other medical fields [12–14]. Despite these advantages, many researchers question the true benefit of these methodologies, as they rightly point out that large amounts of data are usually required to yield accurate results with modern ML algorithms [15]. However, the performance was only measured in terms of area under the curve (AUC), a metric that can be misleading to test the performance of a model [16].

Interpretability of ML results has also been a source of concern for medical researchers, as in many cases, complex data analyses need to be performed requiring high-level programming skills and requiring complex interpretations. However, the implementation of automated ML in easy packages have emerged in recent years helping non-experts to use ML off-the-shelf [17].

To our knowledge, no study has addressed whether novel ML algorithms outperform traditional models in limited sample sizes including the discrimination ability of several algorithms in terms of probability class given by the model. This study presents real-life data of early-treated children with HIV. In this study, we did not aim to explain or describe the course of infection in these children but to select the best predicting algorithm.

## Materials & methods

### Study population

This analysis was performed within an international prospective multicenter Early Antiretroviral Treatment in HIV Children (EARTH) Cohort, embedded in the EPIICAL Consortium (Early treated Perinatally HIV Infected individuals: Improving Children's Actual Life Project).

Patients were recruited from South Africa (Africa Health Research Institute (AHRI), Durban; Family Centre For Research With Ubuntu (FAMCRU), Stellenbosch; Perinatal HIV Research Unit (PHRU), Soweto) and Mozambique (Centro de Investigación en Salud de Manhiça (CISM)–ISGlobal, Manhiça; Fundação Ariel Glaser contra o SIDA Pediatrico, Mozambique). Inclusion criteria included infants living with perinatally acquired HIV who begin ART before 3 months of age, or breastfed infants diagnosed with HIV under three months of age that started ART within three months of diagnosis were enrolled in the study.

In this analysis, we included a total of 100 children with a minimum follow-up time of 12 months and had data on viral load (VL) or percentage of CD4 T-cells (%CD4) available at enrollment until May 1st 2021. Infants and mothers' demographic, clinical, virological, and immunological variables at enrollment were included as possible predictors in the dataset. The combined primary outcome was mortality or clinical progression to AIDS 12 months after enrollment. Clinical progression to AIDS was defined as having a clinical WHO AIDS stage III or IV events after enrollment visit.

The variables included as predictors were the same for all the models: age at recruitment, sex, weight-for-age (WAZ) z-score at enrollment calculated using the WHO Growth Reference [18], preterm birth, age at HIV diagnosis, age at ART initiation, initiation ART regimen, pre-ART VL, baseline %CD4, mother severe life events (including change in employment, separation or relationship break-up, new partner, loss of home or move, or death in the family) or health issues, mother's adherence to ART at enrollment, and mother's last CD4 count and VL measurements.

The EARTH study was approved by ethics committees within each country and written informed consent was obtained from the parent(s)/legal guardian following country-specific law. Each participant received a unique study number, under which data were pseudo-anonymized.

## Models

### Logistic Regression (LR)

This algorithm is considered an extension of linear regression, used to model a dichotomous variable that usually represents the presence or absence of an event. In other words, LR can predict the likelihood of occurrence of a certain outcome [19].

Random Forest (RF): The RF algorithm is built by many decision trees, as a forest is formed by a collection of many trees. To classify a new sample, the input vector of that sample goes through all the decision trees of the forest. Each decision tree gives a classification outcome, i.e., a vote. After a large number of trees is generated, the forest chooses the majority vote for the most popular class and chooses the classification outcome [20].

### Support Vector Machine (SVM)

The SVM technique constructs an optimal hyperplane. In other words, a delimitation to separate the training set according to the outcome into an n-dimensional space, where n is the number of features. The distance between this hyperplane and the nearest instance is the marginal distance for a class. The optimal hyperplane is the one that maximizes this distance for both classes. When the data are not linearly separable, the hyperplane is constructed by a non-linear function (kernel) [21]. For this research, the radial basis kernel was used.

### Naïve Bayes (NB)

The NB algorithm is based on the probabilistic Bayes' theorem. This classifier calculates the probability of an event based on observed values of relevant variables and prior knowledge on

the frequency at which these values occur when the event takes place. The algorithm assumes that each of the variables contributes independently to that probability. The final classification is produced by combining the prior knowledge and observations of predictor variables and comparing the likelihood of an event occurring.

### K-nearest neighbour (KNN)

The KNN algorithm can be considered a variant of NB classifier. Unlike NB, the KNN algorithm does not require prior probabilities. This algorithm assumes that similar events exist in close proximity. In other words, the values of their predictor variables are similar. The implementation of this algorithm is preceded by a clustering technique, used to determine this similarity and therefore the proximity of a new event to a known event.

### Artificial Neural Networks (ANN)

The ANNs are machine learning algorithms inspired by the neural activity of the brain [22]. ANNs are represented by nodes (analogous of neurons) grouped in matrices called layers, which interconnect to other layers to generate knowledge, just as neuronal layers in the brain. Nodes and edges (connections) have weights that are adjusted according to the frequency in which the connections are made, allowing the model (analogous of the brain itself) to recognize patterns in future datasets characteristic of a given class.

### Elastic Net (EN)

The EN is the most recent and optimized penalized regression algorithm used in the machine learning field. These algorithms are based on simple linear regression. However, they 'penalize' the least informative variables by reducing or even eliminating their coefficients, providing a simple and more generalizable model, which helps to reduce the dimensions of the model and its tendency to overfit training data [23].

The caret R package [24] was used to implement stats [25] for LR, randomForest [20] for RF, kernlab [26] for SVM, klaR [27] for NB, nnet [28] package for ANN, and the glmnet [23] package for the Elastic Net.

### Training and validation of the models

All the included participants were randomly divided into a training dataset used to generate the models (70% of the original dataset), and a validation dataset used to assess the performance of the models (remaining 30%). Partitions were balanced by the outcome class, which yielded a training set with an n = 71, and a validation set with an n = 29. Both the training and validation data set was compared to ensure that the validation set was representative of the whole population. Chi-squared tests were used to compare categorical variables and Mann-Whitney U test to compare continuous variables. Absolute numbers and frequencies were assessed for the categorical variables and medians and interquartile ranges [IQR] for the continuous.

For each of the novel 6 ML models, a parameter tuning was performed in the training set with 10 tune grids to select the best-performing hyperparameters of each model using 5 times repeated 10-fold cross-validation. The tuning parameters were the ones implemented in the caret R package (RF: mtry; SVM: C and sigma; NB: fL and adjust; KNN: K; ANN: size and decay; and GLMnet: alpha and lambda). To deal with class imbalance, a down-sampling was performed, in other words, randomly sub-setting all the classes in the training set so that their

class frequencies match the least prevalent class. A p-value <0·05 was taken as statistically significant.

A confusion matrix was built for each of the models to assess their accuracy, sensitivity, and specificity. The area under the curve (AUC) of the receiver operating characteristic curves (ROCs) was also determined for each model [29]. Both apparent AUC (determined using the training set) and actual AUC (determined in the validation set) were calculated. We calculated optimism as the difference between the apparent AUC and validated AUC.

All missing values were imputed independently for the training and the testing dataset, Imputation was done by using a non-parametric approach based on Random Forest (RF) algorithm. The Out-of-bag (OOB) imputation error estimate was assessed as the normalized root mean squared error (NRMSE) and the proportion of falsely classified (PFC) outcomes.

All the analyses were performed using the R language. The caret R package [24] was used to implement the splitting of the data set, model parameter tuning, model training, and confusion matrix validation.

Data augmentation was done to increase the amount of data by adding synthetic samples (n = 280) to the real data that we had (n = 100). From this oversampling data (n = 380), we resampled with replacement three subsets of n = 100 (A1, A2, and A3). The seven models were trained and validated as described above in the A1, A2, and A3 subsets. The data augmentation was performed using the augmenteR R package.

The R source code are publicly available at doi:10.5281/zenodo.6303556.

## Results

A total of 100 children living with perinatally acquired HIV who received early treatment were included in the study. Of those, 33/100 (33%) died or clinically progressed to AIDS within 12-months of follow-up. Specifically, 22% died, 14% progressed, and 3% progressed and then died. The description of the infants' features according to the primary outcome is described in the S1 Table.

The distribution of each randomly divided data set (training and testing) was compared in Table 1. Training and testing data sets did not differ significantly across the variables included in the models. In the imputation of the training data set, the NRMSE was 0·22 for continuous and the PFC 0·14 for categorical, and for the testing, data set NRMSE was 0·01 and the PFC 0·0001. We did not find differences between the imputed dataset and the complete case data set (S2 Table). The tuning parameters for training each of the algorithms tested are described in Table 2.

RF and SVM presented higher accuracy than the rest of the algorithms (RF: 82·8%, SVM: 82·8%, KNN:79·3%, LR: 75·9%, GLMnet: 69%, ANN: 69%, NB:65·5%). RF was the most sensitive algorithm (RF:78%, SVM: 66·7%, KNN:66·7%, ANN: 66·7%, NB: 55·6%, LR:55·6%, and GLMnet: 33.3%). Most algorithms presented similar specificity, except ANN and NB (SVM:90%, RF:85%, GLMnet:85%, LR: 85%, ANN: 70%, and NB: 70%). SVM and RF presented the highest PPV in the validation (SVM: 75%, RF:70%). Likewise, RF (89·5%) and SVM (85·7%) presented the highest NPV in the validation (Fig 1). Complete case analysis was performed, and RF was also the best performing model. The variable importance of each algorithm is summarized in S1 Fig. All the models agree on the baseline VL, age at HIV diagnosis, and WAZ as the most important predictors.

The RF was the algorithm with the largest AUC of the ROC curve (73.2% (67·2–79·1)) followed by the KNN algorithm (69·6% (36·2–76·1)) (Fig 2). The AUC optimism was higher in RF (0.2), LR (0.19), and NB (0.18) than in the rest of the algorithms like KNN (0.05), ANN (0.03), and SVM (0.02). RF-based model was also the one that performed the best in the three synthetic data sets (S2 Table).

**Table 1. Feature distribution according to the different data sets.**

| | Training set | Testing set | p-value |
|---|---|---|---|
| | N = 71 | N = 29 | |
| **Age at recruitment** | | | 0.316 |
| *Days (median, IQR)* | 36.0 [29.6;69] | 41.0 [30;89.1] | |
| **Gender** | | | 0.416 |
| Female | 35 (49.3%) | 11 (37.9%) | |
| Male | 36 (50.7%) | 18 (62.1%) | |
| **Weight-for-age at enrollment** | | | 0.805 |
| *z-score (median, IQR)* | -1.46 [-2.62;-0.87] | -1.18 [-2.98;-0.30] | |
| **Preterm birth** | | | 0.140 |
| No | 41 (57.7%) | 22 (75.9%) | |
| Yes | 30 (42.3%) | 7 (24.1%) | |
| **Age at HIV diagnosis** | | | 0.563 |
| *Days (median, IQR)* | 30.0 [0.00;35.5] | 31.0 [0.00;50.0] | |
| **Age at ART** | | | 0.195 |
| *(median, IQR)* | 32.0 [18.5;62.5] | 36.0 [23.0;82.0] | |
| **Initial ART regimen** | | | 0.558 |
| 3TC+ABC+LPVr | 33 (46.5%) | 14 (48.3%) | |
| 3TC+ABC+NVP | 0 (0.00%) | 1 (3.45%) | |
| 3TC+AZT+LPVr | 22 (31.0%) | 8 (27.6%) | |
| 3TC+AZT+NVP | 16 (22.5%) | 6 (20.7%) | |
| **Baseline viral load** | | | 0.350 |
| *Copies/mL (median, IQR)* | 609715 [36738;2570245] | 226844 [36295;1344319] | |
| **Baseline % CD4** | | | 0.587 |
| *Cell/mL (median, IQR)* | 36.9 [29.9;45.2] | 40.0 [28.0;47.0] | |
| **Maternal severe life events or health issues** | | | 1.000 |
| No | 34 (47.9%) | 14 (48.3%) | |
| Yes | 37 (52.1%) | 15 (51.7%) | |
| **Maternal adherence (self-reported at enrollment)** | | | 0.589 |
| Poor | 3 (4.23%) | 3 (10.3%) | |
| Intermediate low | 12 (16.9%) | 4 (13.8%) | |
| Intermediate high | 18 (25.4%) | 5 (17.2%) | |
| Good | 38 (53.5%) | 17 (58.6%) | |

ART: Antiretroviral; 3TC: Lamivudine; ABC: Abacavir; LPVr: Lopinavir boosted with ritonavir; NVP: Nevirapine; Maternal severe life events: change in employment, separation or relationship break-up, new partner, loss of home or move, or death in the family; Maternal adherence (Optimal: No ART dose missed; Intermediate low: 10–50% doses missed; Intermediate high: 50–90%; Good: >90%)

The probability of having the primary combined outcome calculated by each of the algorithms is described in Fig 3. LR and NB algorithms failed to assess significantly different probabilities for each of the outcome results. RF, SVM, KNN, ANN, and GLMnet were able to assess significantly different outcome probabilities.

## Discussion

The results obtained in this study indicate that the outcome of children living with perinatally acquired HIV who received early treatment can be predicted with considerable accuracy (>80%) by using novel ML algorithms that can integrate clinical, virological, and

**Table 2. Algorithm tuning parameters.**

| Algorithm | Tuning parameter |
|---|---|
| Logistic regression | - |
| Random forest | mtry = 12 |
| Support Vector Machine | C = 8; sigma = $4.69 \cdot 10^{-11}$ |
| Naïve Bayes | fL = 0; adjust = 1 |
| K-nearest neighbor | K = 5 |
| Artificial Neural Network | Size = 11; decay = 0.1 |
| GLMNET | Alpha = 0.8; lambda = 0.21 |

Algorithm tuning parameters selected by repeated (5 times) 10-fold cross-validation in a grid. Mtry: Number of variables for splitting at each tree node in a random forest; C: regularization parameter that controls the trade off between the achieving a low training error and a low testing error; sigma: determines how fast the similarity metric goes to zero as they are further apart; fL: Laplace smoother; adjust: adjust the bandwidth of the kernel density; K = number of nearest neighbours; size: number of units in hidden layer; decay: regularization parameter to avoid over-fitting; alpha: regularization parameter; lambda: penalty on the coefficients

immunological data. The RF algorithm outperformed in sensitivity all other tested ML methods, including LR, commonly used in the medical field, even trained in a small sample size (n = 71).

Due to its flexibility, ML is broadly considered to be a better analytical tool than traditional statistical modelling, such as logistic regression. One of the main reasons why this may be is due to ML can overcome issues of multiple and correlated predictors, non-linear relationships, and interactions between predictors and endpoints that can lead to misassumptions and hyperparametric scenarios in traditional regression models [30]. However, LR has been the standard of care for binary classification. This is mainly because LR does not demand computational or statistical expertise and also because novel ML are believed to be data-hungry [15].

The limited sample size hurdle was very present in our study since acquiring hundreds or thousands of patients' samples in some fields such as pediatric HIV is rarely feasible, particularly in the case where a follow-up is needed in countries with limited resources. We have shown that ML learning approaches including RF, SVM, ANN, or KNN can offer a performance advantage over traditional LR methods even in small sample sizes in terms of accuracy, sensitivity, and specificity. Of special interest, our results have shown that the RF model yielded significantly different outcome probabilities for each class unlike the other models applied. This means that the outcome probabilities given by this model could help the physicians to identify those children who will die or progress better than with the standard LR. These findings agree with other studies where RF performed better than LR [12, 14].

The AUC optimism given by the models were higher in all the models due to the limited sample size. Also, the RF, LR and NB models presented the highest differences between AUC validated and apparent AUC. However, in our study, LR did not present the lowest optimism as reported in other studies [15]. The data hungriness or performance should not be only evaluated with the AUC metric [16], which would be not the best measurement for all prediction goals including, for example, where maximizing sensitivity could save patients' lives. We believe that with small data any algorithm will be penalized in robustness and the performance metric should be evaluated depending on the outcome of interest. In the clinical field, there are complex associations and highly correlated data, issues that RF copes better than traditional methods.

Interpretability of ML results has also been a source of concern for medical researchers, as in many cases, the complex data analyses performed do not yield clear and simple relationships

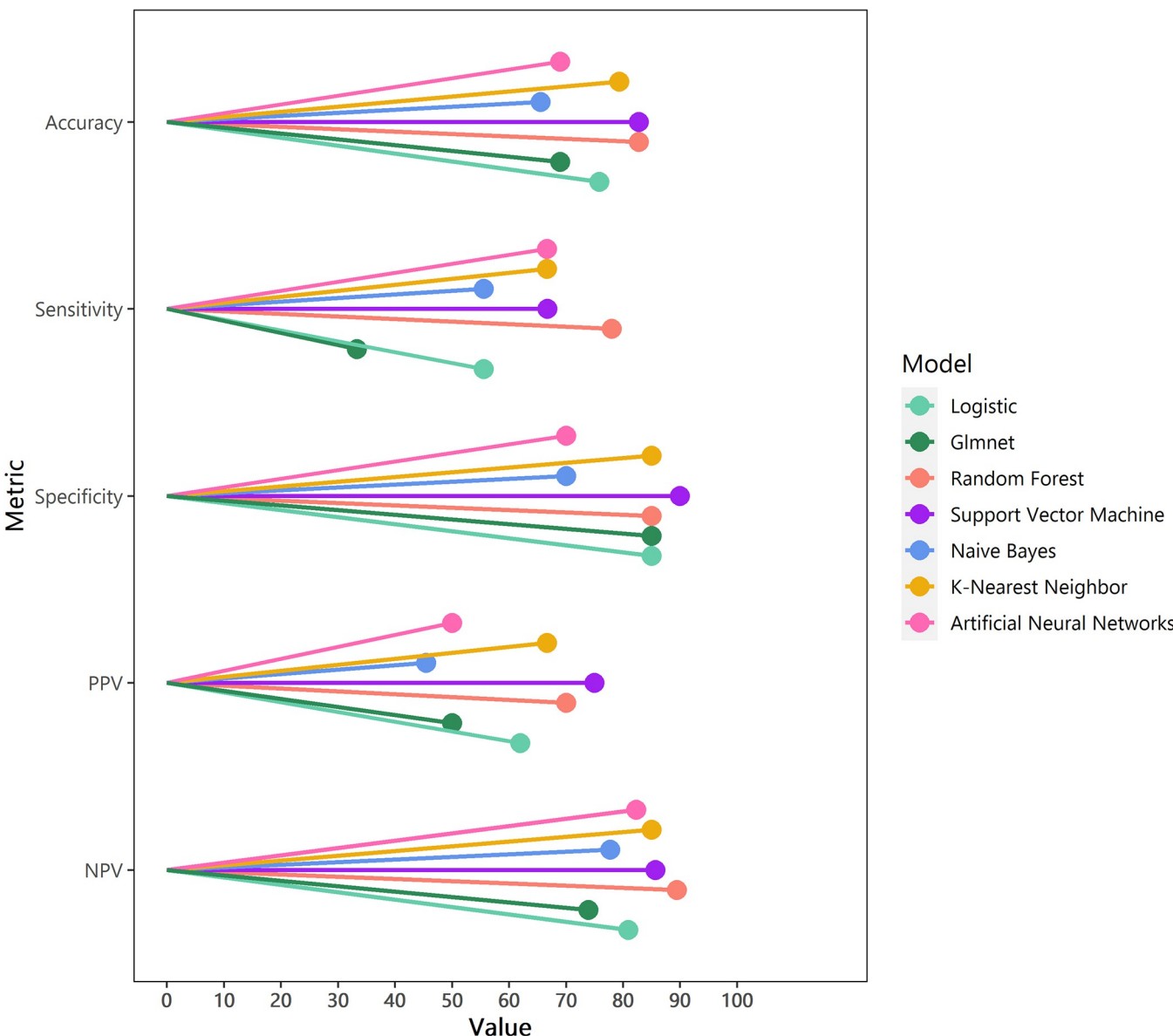

**Fig 1. Algorithms performance in the validation set.**

between variables and outcomes. Therefore, we believe that traditional LR may be the most suitable model for studies that aim to not just predict or classify outcomes, but, rather, to measure associations between specific factors and an event.

In limited resources countries where HIV patients' care is decentralized and no specialized staff is attending children, accurate models are needed to alert the clinician if the patient is at risk of clinical progression. In this case, clinicians may consider referring the patients to a higher level of care or a different follow-up schedule.

In terms of limitations, the main drawback of this research is the limited number of observations, which likely diminishes the prediction capacity of all the trained models, designed to work with a much larger number of events. The design where the training and validation set

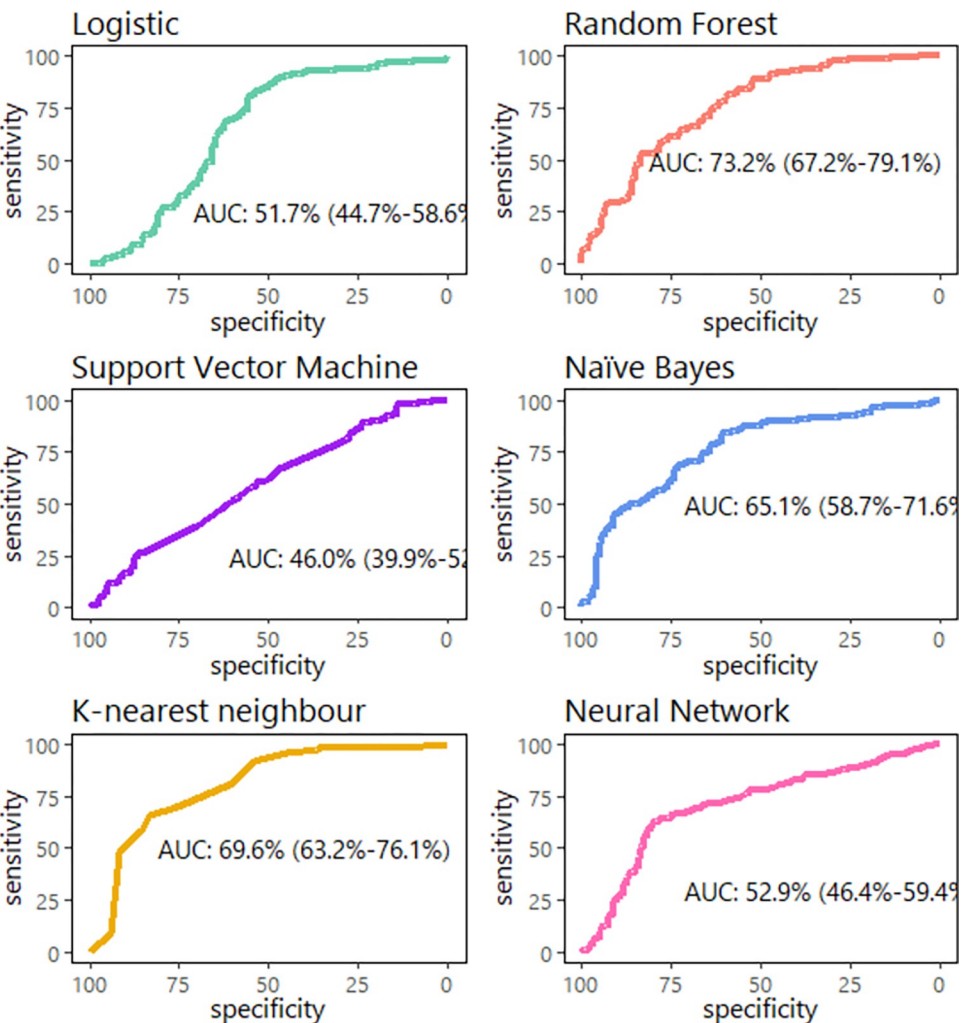

**Fig 2. Algorithms receiving operating curve in the validation set.**

are separated could exacerbate the limitation of sample size. This could be solved with methods like nested cross-validations in which the validation set also remained completely blind for the model. However, we decided to use a unique built-in R package as caret to reduce programming complexity and these novel methods are not implemented automatically. Further, we employed the machine learning classifiers without extensive optimizations, leading to the possibility that our results could be improved by performing these adjustments. Because of the same reason, we decided to follow the simplest way to implement modern ML. To make more robust the comparison of the models in small datasets, three additional synthetic small data sets (n = 100) were created to perform the same comparisons. Results were consistent among them, and RF was the best fitting model in the real and the synthetic datasets.

The present study has several strengths. This study has brought ML close to the physicians by designing a digital comprehensive friendly app with direct applicability to healthcare. The collaboration of multidisciplinary staff such as physicians, bioinformaticians and biostatistician is of great value in identifying new and more efficient statistical methods to better interpret clinical data and aid clinical practice/patient management.

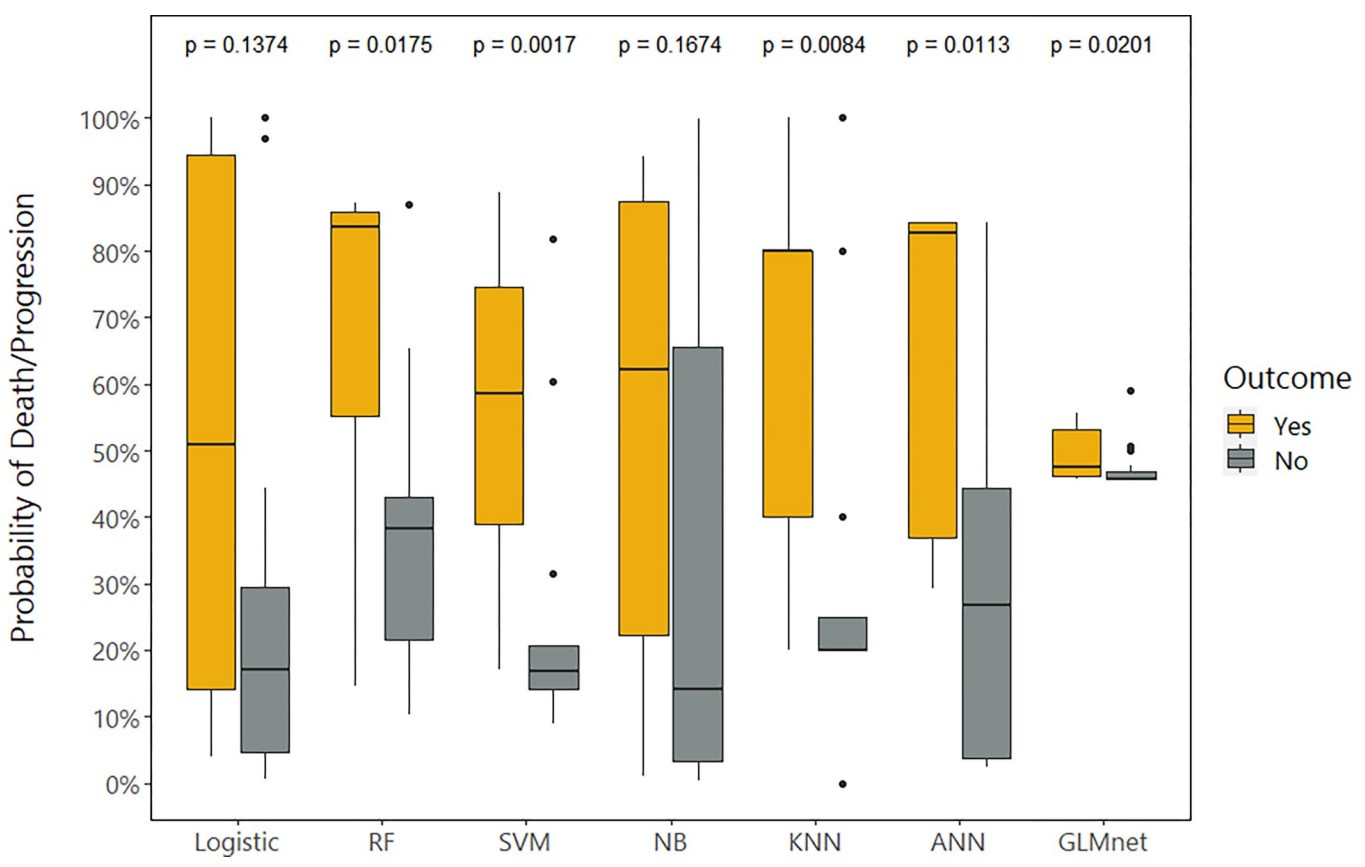

**Fig 3. Probability of death/progression according to each algorithm in the validation set.**

## Conclusions

Despite early treatment and close follow-up, death and progression remain unacceptably high in infants living with HIV in Africa, especially in the first 6 months of life. This study shows how infant mortality and progression to AIDS in early-treated HIV-infected infants can be predicted more accurately with modern ML algorithms such as RF than with traditional LR. These state-of-the-art ML algorithms outperformed common LR also in a small sample size. This result is particularly interesting since recruitment difficulties and ethical considerations make pediatric studies more challenging, especially with critically ill children. In such cases, the importance of accurate models is stressed.

## Supporting information

**S1 Table. Study population characteristics according to the primary outcome (Death/clinical progression to AIDS).**
(DOCX)

**S2 Table. Feature distribution in the original and imputed data sets.**
(DOCX)

**S3 Table. Model performance in the different subsets.**
(DOCX)

**S1 Fig. Variable importance according to each algorithm.**
(TIF)

**S1 Questionnaire. Inclusivity in global research.**
(DOCX)

## Acknowledgments

We thank the EPIICAL Consortium team, all the staff involved in the EARTH cohort, and thank the families and participants who participate in the study. We also want to thank Angel Panizo for the artificial intelligent support.

Members of the EPIICAL Consortium are Paolo Rossi, Carlo Giaquinto, Silvia Faggion, Daniel Gomez Pena, Inger Lindfors Rossi, William James, Alessandra Nardone, Paolo Palma, Paola Zangari, Carla Paganin, Eleni Nastouli, Moira Spyer, Anne-Genevieve Marcelin, Vincent Calvez, Pablo Rojo, Alfredo Tagarro, Sara Dominguez, Maria Angeles Munoz, Caroline Foster, Savita Pahwa, Anita De Rossi, Mark Cotton, Nigel Klein, Deborah Persaud, Rob J De Boer, Juliane Schroeter, Adriana Ceci, Viviana Giannuzzi, Kathrine Luzuriaga, Nicolas Chomont, Nicola Cotugno, Paolo Rossi, Louise Kuhn, Andrew Yates, Avy Violari, Kennedy Otwombe, Paula Vaz, Maria Grazia Lain, Elisa López-Varela, Tacilta Nhamposssa, Elisa Lopez, Denise Naniche, Ofer Levy, Philip Goulder, Mathias Lichterfeld, Holly Peay, Pr Mariam Sylla, Almoustapha Maiga, Cissy Kityo, Thanyawee Puthanakit.

## Author Contributions

**Conceptualization:** Sara Domínguez-Rodríguez, Shaun Barnabas, Pablo Rojo, Alfredo Tagarro.

**Data curation:** Sara Domínguez-Rodríguez, Miquel Serna-Pascual, Andrea Oletto.

**Formal analysis:** Sara Domínguez-Rodríguez.

**Funding acquisition:** Paolo Rossi, Carlo Giaquinto.

**Investigation:** Sara Domínguez-Rodríguez, Peter Zuidewind, Els Dobbels, Siva Danaviah, Osee Behuhuma, Maria Grazia Lain, Paula Vaz, Sheila Fernández-Luis, Tacilta Nhampossa, Elisa Lopez-Varela, Kennedy Otwombe, Afaaf Liberty, Avy Violari, Almoustapha Issiaka Maiga, Louise Kuhn, Pablo Rojo.

**Methodology:** Sara Domínguez-Rodríguez.

**Project administration:** Paolo Rossi, Carlo Giaquinto.

**Resources:** Shaun Barnabas, Peter Zuidewind, Els Dobbels, Siva Danaviah, Osee Behuhuma, Maria Grazia Lain, Paula Vaz, Sheila Fernández-Luis, Tacilta Nhampossa, Elisa Lopez-Varela, Kennedy Otwombe, Afaaf Liberty, Avy Violari, Almoustapha Issiaka Maiga, Carlo Giaquinto.

**Supervision:** Paolo Rossi, Carlo Giaquinto, Pablo Rojo, Alfredo Tagarro.

**Writing – original draft:** Sara Domínguez-Rodríguez, Pablo Rojo, Alfredo Tagarro.

**Writing – review & editing:** Andrea Oletto, Shaun Barnabas, Peter Zuidewind, Els Dobbels, Siva Danaviah, Osee Behuhuma, Maria Grazia Lain, Paula Vaz, Sheila Fernández-Luis, Tacilta Nhampossa, Elisa Lopez-Varela, Kennedy Otwombe, Afaaf Liberty, Avy Violari, Almoustapha Issiaka Maiga, Paolo Rossi, Carlo Giaquinto, Louise Kuhn.

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
