## [Decision Letter · Decision Letter 0]

18 Jan 2022

PONE-D-21-31247Machine learning outperformed logistic regression classification even with limit sample size: A model to predict pediatric HIV mortality and clinical progression to AIDSPLOS ONE

Dear Dr. Tagarro,

Thank you for submitting your manuscript to PLOS ONE. After careful consideration, we feel that it has merit but does not fully meet PLOS ONE’s publication criteria as it currently stands. Therefore, we invite you to submit a revised version of the manuscript that addresses the points raised during the review process.

We look forward to receiving your revised manuscript.

Kind regards,

Zaher Mundher Yaseen

Academic Editor

PLOS ONE

Journal Requirements:

This work has been supported within EPIICAL project by through an independent grant to the PENTA (Paediatric European Network for Treatment of AIDS) Foundation. The funders had no role in study design, data collection, analysis, and interpretation, or manuscript preparation.

6. Please note that in order to use the direct billing option the corresponding author must be affiliated with the chosen institute. Please either amend your manuscript to change the affiliation or corresponding author, or email us at plosone@plos.org with a request to remove this option.

7. We note that you have stated that you will provide repository information for your data at acceptance. Should your manuscript be accepted for publication, we will hold it until you provide the relevant accession numbers or DOIs necessary to access your data. If you wish to make changes to your Data Availability statement, please describe these changes in your cover letter and we will update your Data Availability statement to reflect the information you provide.

8. PLOS requires an ORCID iD for the corresponding author in Editorial Manager on papers submitted after December 6th, 2016. Please ensure that you have an ORCID iD and that it is validated in Editorial Manager. To do this, go to ‘Update my Information’ (in the upper left-hand corner of the main menu), and click on the Fetch/Validate link next to the ORCID field. This will take you to the ORCID site and allow you to create a new iD or authenticate a pre-existing iD in Editorial Manager. Please see the following video for instructions on linking an ORCID iD to your Editorial Manager account: https://www.youtube.com/watch?v=_xcclfuvtxQ

9. One of the noted authors is a group or consortium [EPIICAL Consortium]. In addition to naming the author group, please list the individual authors and affiliations within this group in the acknowledgments section of your manuscript. Please also indicate clearly a lead author for this group along with a contact email address.

Reviewers' comments:

Reviewer's Responses to Questions

**Comments to the Author**

1. Is the manuscript technically sound, and do the data support the conclusions?

Reviewer #1: Yes

Reviewer #2: Yes

Reviewer #3: Partly

2. Has the statistical analysis been performed appropriately and rigorously? 

Reviewer #1: Yes

Reviewer #2: Yes

Reviewer #3: I Don't Know

3. Have the authors made all data underlying the findings in their manuscript fully available?

Reviewer #1: Yes

Reviewer #2: Yes

Reviewer #3: No

4. Is the manuscript presented in an intelligible fashion and written in standard English?

Reviewer #1: Yes

Reviewer #2: Yes

Reviewer #3: Yes

5. Review Comments to the Author

Reviewer #1: Thank you for asking my opinion about the manuscript entitled ‎"Machine learning outperformed logistic regression classification even with limit sample size: A model to predict pediatric HIV mortality and clinical progression to AIDS" This manuscript ‎aims to compare the performance of 7 algorithms in the prediction of 1-year mortality and clinical progression to AIDS in a small cohort of infants living with HIV from South Africa and Mozambique. I believe that this manuscript should ‎be minor revision:‎

The aim of this study (Page 1- Abstract) was there explore compare the performance of 7 algorithms in the prediction of 1-year mortality and clinical progression to AIDS in a small cohort of infants living with HIV from South Africa and Mozambique. There were several good things about the paper, such as ‎aim good. But the abstract should be reformulated and the ‎objective of the study should be well highlighted.‎

‎1. The abstract is very long.‎

‎2. In the introduction, include the significance of the study as ‎well as novelty. What makes the study different from the rest and ‎what does it add to the current knowledge?‎

‎2.1. In the introduction, the authors should have explained the ‎purpose of this study and the existing gaps in this field and ‎explain why this study was conducted.‎

‎3. In the material and methods section, include sample size justification. ‎

‎4. Conclusions are too short and not supported by results.‎

‎5. References are relevant, correct, and not recent. The number of ‎references should be increased.‎

‎6. There are a lot of grammatical errors. This must be taken care ‎of and addressed.‎

Reviewer #2: I find out this article very interesting. It construction is very good and language fulfills scientific research standards.

Selected methods from ML for comparison are general enough and popular enough. I recommend this article for publication.

However I want to authors check if the the sigma value in the SVM method is accurate. If the sigma is very small the SVM tends to make a local classifier, larger sigma tends to make a much more general classifier. Please check if the optimization of this parameter was done right.

The second thing, which will make this article more clearer to the reader, would be strict references to the Project R packages. I am familiar with this software, but I think that if someone is not, it would be hard to him to reproduce the methods used in results discussion.

Reviewer #3: Summary:

The authors compared seven well-understood machine learning (ML) approaches to the problem of predicting mortality and progression to full AIDS on infants with HIV from South Africa and Mozambique. They compared prediction accuracy, sensitivity, and specificity of these methods to highlight the relative benefits of some ML methods and especially against logistic regression, a very common prediction method in medicine. Developing a more accurate predictive model for health outcomes of HIV-infected infants is a very worthwhile task.

Technical Soundness:

The largest obvious challenge with this study, and one that the authors openly recognize, is the small dataset to be used for training and validation. With a 70/30 test-train split on n=100 observations, I believe the authors are challenged to convincingly demonstrate the generalizability of their comparative study. Some ML methods intrinsically perform better than others with smaller training datasets (e.g. Random Forest vs. ANN), but I remain unconvinced the the relative performance of the algorithms described herein would still hold true if one could train on more data. In other words, are the reported relative results really indicative of the true efficacy of the methods or are they significantly influenced by data sparsity alone?

I think it would benefit this comparative study to increase the number of observations for training and validation by at least an order of magnitude. If that can't be done in a practical way with actual observational data due to real-world constraints, I would recommend that the authors look at tabular data augmentation methods to synthetically increase the size of the training and test sets. Tabular data augmentation such as Test-Time augmentation (TTA) can synthetically add new observations by adding small amounts of gaussian noise to duplicated data rows. Libraries such as DeltaPy (https://github.com/firmai/deltapy) also assist with meaningful tabular data augmentation and Generative adversarial networks (GANs) and other methods have also been used for this (https://arxiv.org/abs/2010.00638). I request the authors to find a way to train and test their algorithms on more data to at least quantify/estimate the effect of small dataset size on the relative performance of the tested results.

I would have liked to see some attempt at feature importance estimation under all 7 different algorithms. Which independent variables have the most influence on the balanced accuracy of various ML models? While a bit more challenging for some methods such as ANN, determining relative feature importance under logistic regression and Random Forest is direct and straightforward. Even with ANN, you should be able to do Permutation Feature Importance estimation. Ideally, I would like to see a table of algorithms vs. features/variables ranked by importance. In other words, which features contribute the most to prediction accuracy across the various ML methods? And is there consistency in feature importance across ML methods?

Figure 2 shows the AUC under the ROC curve and is very telling. An AUC of 50% indicates a totally random classifier, and several of the reviewed algorithms approximated a random classifier with an AUC of close to 50%: Logistic (51.7% AUC), SVM (46% AUC), ANN (52.9% AUC). Really, only Random Forest performed acceptably (73.2%) by the AUC metric, probably because RF had the highest sensitivity. Still, an AUC of 73% is not great. I would love to see how these AUC might change/improve as a function of additional training data.

Given the known class imbalance in the data, the authors chose to randomly subsample the already limited training data so that the class frequencies matched the least prevalent class. Another way around this might have been to use a confusion matrix, but report balanced accuracy [(sensitivity + specificity)/2] instead of simple accuracy.

Table 2 shows the tuned hyperparameters discovered by doing a gridded 10-fold cross validation. These are not comprehensive hyperparameters for these algorithms, so it would be helpful to understand how these were chosen. For example, did you only optimize mtry for RF? How did you chose this hyper parameter instead of ntree, sampsize, maxnodes, etc.? Can you include justification for which hyperparameters you chose to optimize for each method?

The data is well-described in the manuscript, but the data is not publicly available. The authors stated that the data is protected by GDPR. I suspect that a sufficiently anonymized version of these data could have been made public in a way that did not conflict with GDPR. If not, please include the actual URL or DOI to the REDCap instance holding these data with better instructions for requesting access.

It would be helpful to review your R source code. Can the authors please publish this source code, perhaps in a public GitHub repository and share the URL or DOI to access to the source code?

Minor editing: There are various misspellings such as "enrolment" instead of enrollment on lines 46-51. Or "tunning" instead of tuning in Table 2. Lines 181-184 have some english word choice problems such as "These" instead of "This" or "will death" instead of "will die".

6. PLOS authors have the option to publish the peer review history of their article (what does this mean?). If published, this will include your full peer review and any attached files.

Reviewer #1: No

Reviewer #2: No

Reviewer #3: **Yes: **Luke Sheneman

---

## [Author Response · Author response to Decision Letter 0]

21 Jul 2022

Respond to each of the reviewers have been attached as respond to reviewers in the general information to be attached. In that document you can see the answer highlighted in other color to facilitate the reading

Editor comments: Journal requirements

We have reformatted the manuscript according to formatting guidelines

2. Please include a complete copy of PLOS’ questionnaire on inclusivity in global research in your revised manuscript. Our policy for research in this area aims to improve transparency in the reporting of research performed outside of researchers’ own country or community. The policy applies to researchers who have travelled to a different country to conduct research, research with Indigenous populations or their lands, and research on cultural artefacts. The questionnaire can also be requested at the journal’s discretion for any other submissions, even if these conditions are not met. Please find more information on the policy and a link to download a blank copy of the questionnaire here: https://journals.plos.org/plosone/s/best-practices-in-research-reporting. Please upload a completed version of your questionnaire as Supporting Information when you resubmit your manuscript. A copy of PLOS questionnaire on inclusivity in global research has been attached. 

We have included additional information about participant informed consent in the Methods section, study population subsection. 

4. We note that the grant information you provided in the ‘Funding Information’ and ‘Financial Disclosure’ sections do not match. We have included the funder ViiV Healthcare in the cover letter 

We have included the funding information in the cover letter and remove the funding statement from the acknowledgements section. 

6. Please note that in order to use the direct billing option the corresponding author must be affiliated with the chosen institute. Please either amend your manuscript to change the affiliation or corresponding author, or email us at plosone@plos.org with a request to remove this option. The affiliation of the corresponding author has been modified in the manuscript

7. We note that you have stated that you will provide repository information for your data at acceptance. Should your manuscript be accepted for publication, we will hold it until you provide the relevant accession numbers or DOIs necessary to access your data. If you wish to make changes to your Data Availability statement, please describe these changes in your cover letter and we will update your Data Availability statement to reflect the information you provide. The data availability statement has been included in the cover letter.

8. PLOS requires an ORCID iD for the corresponding author in Editorial Manager on papers submitted after December 6th, 2016. Please ensure that you have an ORCID iD and that it is validated in Editorial Manager. To do this, go to ‘Update my Information’ (in the upper left-hand corner of the main menu), and click on the Fetch/Validate link next to the ORCID field. This will take you to the ORCID site and allow you to create a new iD or authenticate a pre-existing iD in Editorial Manager. Please see the following video for instructions on linking an ORCID iD to your Editorial Manager account: https://www.youtube.com/watch?v=_xcclfuvtxQ

The ORCID ID from the corresponding author has been updated

9. One of the noted authors is a group or consortium [EPIICAL Consortium]. In addition to naming the author group, please list the individual authors and affiliations within this group in the acknowledgments section of your manuscript. Please also indicate clearly a lead author for this group along with a contact email address.

The members of the EPIICAL consortium have been listed in the Acknowledgements section

Review Comments to the Author

Reviewer #1: Thank you for asking my opinion about the manuscript entitled ‎"Machine learning outperformed logistic regression classification even with limit sample size: A model to predict pediatric HIV mortality and clinical progression to AIDS" This manuscript ‎aims to compare the performance of 7 algorithms in the prediction of 1-year mortality and clinical progression to AIDS in a small cohort of infants living with HIV from South Africa and Mozambique. 

I believe that this manuscript should ‎be minor revision:‎

The aim of this study (Page 1- Abstract) was there explore compare the performance of 7 algorithms in the prediction of 1-year mortality and clinical progression to AIDS in a small cohort of infants living with HIV from South Africa and Mozambique. There were several good things about the paper, such as ‎aim good. But the abstract should be reformulated and the ‎objective of the study should be well highlighted.‎

1. The abstract is very long.‎

We agree with the reviewer that the abstract was long. However, there is several technical information on methods that take up space and we cannot exclude them. We have now provided an abstract with less than 250 words

‎2. In the introduction, include the significance of the study as ‎well as novelty. What makes the study different from the rest and ‎what does it add to the current knowledge?‎

‎2.1. In the introduction, the authors should have explained the ‎purpose of this study and the existing gaps in this field and ‎explain why this study was conducted.‎

The gaps existing in this fields are detailed in line 18-28 of the introduction where it is explained that despite the advantages on the application of Machine learning stated in previous paragraph, the usability is scarce because the need of large amount of data and lack of interpretabilities models. We have provided line 29-32 explaining the novelty of the study and the aim and discuss the strengths of the study on the discussion section. We have now included in the text that this study presents real-life data instead of simulated data sets.

‎3. In the material and methods section, include sample size justification. ‎

We agree with the reviewer that a sample size justification will be interesting for the studies to assess how sample size was chosen. In this study we included a total of 100 children with information available at the moment of the analysis. This information has been newly included in line 48 in the study population subsection of the Material and Methods section. For that reason, there is no sample size calculation. In addition, the required sample size for a particular machine learning model is often unknown. Pre hoc approaches for estimating sample size for machine learning algorithms are known to be not robust in high dimensionality medical data and highly depends on the variability of the sample. 

‎4. Conclusions are too short and not supported by results.‎

We have included a more detailed paragraph in the conclusions section of the manuscript 

‎5. References are relevant, correct, and not recent. The number of ‎references should be increased.‎

We have included the most recent literature available at the moment. Following the reviewer advice, we have included some more references. 

‎6. There are a lot of grammatical errors. This must be taken care ‎of and addressed.‎

We have now corrected grammatical errors and manuscript have been reviewed by a native co-author 

Reviewer #2: 

I find out this article very interesting. It construction is very good and language fulfills scientific research standards.

Selected methods from ML for comparison are general enough and popular enough. I recommend this article for publication.

However I want to authors check if the the sigma value in the SVM method is accurate. If the sigma is very small the SVM tends to make a local classifier, larger sigma tends to make a much more general classifier. Please check if the optimization of this parameter was done right.

We agree with the reviewer in the association between tuning parameters of each model and the results. Also happening with SVM method. In this study we aim to compare the seven methods in predicting the probability of a composite endpoint (death and progression) and we want to follow an off-the-shelf solution without in-house customization that could lead to difficulties in interpreting results or reproducibility. For that reason, we performed an automatic parameter tuning implemented in the caret R package. For each of the novel 6 ML models, a parameter tuning was performed in the training set with 10 tune grids to select the best-performing hyperparameters of each model using 5 times repeated 10-fold cross-validation.

We are aware that, as the reviewer suggested, better models could have been fitted using extensive tuning of the parameters. However, we tried to make it easier and reproducible in the field.

The second thing, which will make this article more clearer to the reader, would be strict references to the Project R packages. I am familiar with this software, but I think that if someone is not, it would be hard to him to reproduce the methods used in results discussion.

We have newly included in the methods section the packages and references for them.

Reviewer #3: 

The authors compared seven well-understood machine learning (ML) approaches to the problem of predicting mortality and progression to full AIDS on infants with HIV from South Africa and Mozambique. They compared prediction accuracy, sensitivity, and specificity of these methods to highlight the relative benefits of some ML methods and especially against logistic regression, a very common prediction method in medicine. Developing a more accurate predictive model for health outcomes of HIV-infected infants is a very worthwhile task.

Technical Soundness:

The largest obvious challenge with this study, and one that the authors openly recognize, is the small dataset to be used for training and validation. With a 70/30 test-train split on n=100 observations, I believe the authors are challenged to convincingly demonstrate the generalizability of their comparative study. Some ML methods intrinsically perform better than others with smaller training datasets (e.g. Random Forest vs. ANN), but I remain unconvinced the the relative performance of the algorithms described herein would still hold true if one could train on more data. In other words, are the reported relative results really indicative of the true efficacy of the methods or are they significantly influenced by data sparsity alone?

I think it would benefit this comparative study to increase the number of observations for training and validation by at least an order of magnitude. If that can't be done in a practical way with actual observational data due to real-world constraints, I would recommend that the authors look at tabular data augmentation methods to synthetically increase the size of the training and test sets. Tabular data augmentation such as Test-Time augmentation (TTA) can synthetically add new observations by adding small amounts of gaussian noise to duplicated data rows. Libraries such as DeltaPy (https://github.com/firmai/deltapy) also assist with meaningful tabular data augmentation and Generative adversarial networks (GANs) and other methods have also been used for this (https://arxiv.org/abs/2010.00638). I request the authors to find a way to train and test their algorithms on more data to at least quantify/estimate the effect of small dataset size on the relative performance of the tested results.

We agree with the reviewer that sample size is one of the main limitation of this study. We believe that all of these 7 algorithms, including conventional logistic regression, would benefit from increasing sample size. Our aim was to assess the performance of these 7 algorithms under the same sample size limitation to adjust for real-life research in this setting. We agree that increasing sample size could even make these differences bigger since algorithms such as random forest has already demonstrated better performance in high-dimensional data. The novelty of this approach is to describe and compare the performance of the 7 algorithms under sample size limitation. As the reviewer suggested, we have explored some of the packages for data augmentation such as DeltaPy. However, these approaches are based on data augmentation for deep learning more focused on images classification rather than our aim. In any case, we will explore further types of simulation in next steps and try to find a way to increase sample size based on a sample features distribution.

I would have liked to see some attempt at feature importance estimation under all 7 different algorithms. Which independent variables have the most influence on the balanced accuracy of various ML models? While a bit more challenging for some methods such as ANN, determining relative feature importance under logistic regression and Random Forest is direct and straightforward. Even with ANN, you should be able to do Permutation Feature Importance estimation. Ideally, I would like to see a table of algorithms vs. features/variables ranked by importance. In other words, which features contribute the most to prediction accuracy across the various ML methods? And is there consistency in feature importance across ML methods?

We have now included the Supplementary Figure 1 to describe feature importance across ML the models and the results in Results section.

Figure 2 shows the AUC under the ROC curve and is very telling. An AUC of 50% indicates a totally random classifier, and several of the reviewed algorithms approximated a random classifier with an AUC of close to 50%: Logistic (51.7% AUC), SVM (46% AUC), ANN (52.9% AUC). Really, only Random Forest performed acceptably (73.2%) by the AUC metric, probably because RF had the highest sensitivity. Still, an AUC of 73% is not great. I would love to see how these AUC might change/improve as a function of additional training data.

As mentioned before, we agree with the reviewer that increasing the sample size will improve the performance of each of the models. There are several studies exploring the better performance of ML methods in high dimensional and big data. However, we wanted to compare the performance of these algorithms under the same limitations. 

Given the known class imbalance in the data, the authors chose to randomly subsample the already limited training data so that the class frequencies matched the least prevalent class. Another way around this might have been to use a confusion matrix, but report balanced accuracy [(sensitivity + specificity)/2] instead of simple accuracy.

We agree with the reviewer that we could have modified the cut-off that configures the classifier determining an appropriate threshold to balance the accuracy. However we believe that was better to perform downsampling so that physicians and epidemiologist would understand better the sensitivity and specificity result and be more comparable among the ones reported in this setting medical field.

Table 2 shows the tuned hyperparameters discovered by doing a gridded 10-fold cross validation. These are not comprehensive hyperparameters for these algorithms, so it would be helpful to understand how these were chosen. For example, did you only optimize mtry for RF? How did you chose this hyper parameter instead of ntree, sampsize, maxnodes, etc.? Can you include justification for which hyperparameters you chose to optimize for each method?

The hyper parameters shown in Table 2 are the tuning parameters that could be modified and adjust in the caret R package. Tuning parameters are now described in methods section

The data is well-described in the manuscript, but the data is not publicly available. The authors stated that the data is protected by GDPR. I suspect that a sufficiently anonymized version of these data could have been made public in a way that did not conflict with GDPR. If not, please include the actual URL or DOI to the REDCap instance holding these data with better instructions for requesting access.

The study project is still ongoing and database locking has not been performed yet. In addition, this data is allocated and owned by the PENTA Foundation, in Italy and regulated under European GDPR data protection law. According to this regulation, this data is pseudonymized and not anonymized because birth date and site of recruitment is present. Also, there is not a formal consent from the patients or their caregivers to have their data openly available. However, data could be shared under specific circumstances and/or research collaboration by sending a research proposal to the corresponding author of this article.

It would be helpful to review your R source code. Can the authors please publish this source code, perhaps in a public GitHub repository and share the URL or DOI to access to the source code?

We have included the R code in a public repository. The R source code are publicly available at doi:10.5281/zenodo.6303556.

Minor editing: There are various misspellings such as "enrolment" instead of enrollment on lines 46-51. Or "tunning" instead of tuning in Table 2. Lines 181-184 have some english word choice problems such as "These" instead of "This" or "will death" instead of "will die".

We have now corrected grammatical errors and manuscript have been reviewed by a native co-author

---

## [Decision Letter · Decision Letter 1]

13 Sep 2022

PONE-D-21-31247R1Machine learning outperformed logistic regression classification even with limit sample size: A model to predict pediatric HIV mortality and clinical progression to AIDSPLOS ONE

Dear Dr. Tagarro,

Thank you for submitting your manuscript to PLOS ONE. After careful consideration, we feel that it has merit but does not fully meet PLOS ONE’s publication criteria as it currently stands. Therefore, we invite you to submit a revised version of the manuscript that addresses the points raised during the review process.

 Please submit your revised manuscript by Oct 28 2022 11:59PM. If you will need more time than this to complete your revisions, please reply to this message or contact the journal office at plosone@plos.org. Please include the following items when submitting your revised manuscript:A rebuttal letter that responds to each point raised by the academic editor and reviewer(s). You should upload this letter as a separate file labeled 'Response to Reviewers'.A marked-up copy of your manuscript that highlights changes made to the original version. You should upload this as a separate file labeled 'Revised Manuscript with Track Changes'.An unmarked version of your revised paper without tracked changes. You should upload this as a separate file labeled 'Manuscript'.If applicable, we recommend that you deposit your laboratory protocols in protocols.io to enhance the reproducibility of your results. Protocols.io assigns your protocol its own identifier (DOI) so that it can be cited independently in the future. For instructions see: https://journals.plos.org/plosone/s/submission-guidelines#loc-laboratory-protocols. Additionally, PLOS ONE offers an option for publishing peer-reviewed Lab Protocol articles, which describe protocols hosted on protocols.io. Read more information on sharing protocols at https://plos.org/protocols?utm_medium=editorial-email&utm_source=authorletters&utm_campaign=protocols.

We look forward to receiving your revised manuscript.

Kind regards,

Zaher Mundher Yaseen

Academic Editor

PLOS ONE

Journal Requirements:

Reviewers' comments:

Reviewer's Responses to Questions

**Comments to the Author**

1. If the authors have adequately addressed your comments raised in a previous round of review and you feel that this manuscript is now acceptable for publication, you may indicate that here to bypass the “Comments to the Author” section, enter your conflict of interest statement in the “Confidential to Editor” section, and submit your "Accept" recommendation.

Reviewer #1: All comments have been addressed

Reviewer #3: All comments have been addressed

2. Is the manuscript technically sound, and do the data support the conclusions?

Reviewer #1: Yes

Reviewer #3: Partly

3. Has the statistical analysis been performed appropriately and rigorously? 

Reviewer #1: Yes

Reviewer #3: I Don't Know

4. Have the authors made all data underlying the findings in their manuscript fully available?

Reviewer #1: Yes

Reviewer #3: No

5. Is the manuscript presented in an intelligible fashion and written in standard English?

Reviewer #1: Yes

Reviewer #3: Yes

6. Review Comments to the Author

Reviewer #1: Thanks to the corresponding author, he was successful in ‎answering our questions.

Thank you for asking my opinion about the ‎manuscript ‎entitled ‎‎"Machine learning outperformed logistic regression classification even with limit sample size: A model to predict pediatric HIV mortality and clinical progression to AIDS". This ‎manuscript ‎aims to compare the performance of 7 algorithms in the prediction of 1-year mortality and clinical progression to AIDS in a small cohort of infants living with HIV from South Africa and Mozambique. I believe that this manuscript should ‎be ‎minor revision.

Reviewer #3: The authors have revised their manuscript wherein they compared seven well-understood machine learning (ML) approaches to the problem of predicting mortality and progression to full AIDS on infants with HIV from South Africa and Mozambique. The authors should be congratulated for tackling such meaningful and important work.

The authors responded to all reviewer comments and made corrections or additions where possible to address reviewer feedback.

The largest problem with this study is the size of the dataset (n=100), as the authors fully recognize. It is not convincing to compare/contrast the performance of machine learning methods against only one extremely small dataset. There is not enough evidence here to demonstrate that Random Forest will generally outperform other methods for these kinds of data. All one can say is that RF outperformed other methods for this single, small dataset. That is an interesting datapoint, but I'm afraid I still remain unconvinced due to lack of sample size and/or replicates. If real-world data was not available (for practical reasons), I would have liked to see some kind of data augmentation via simulation to increase the length of this dataset OR simulate a number of small synthetic n=100 datasets with similar feature distributions. The authors mentioned that their goal was to compare ML methods specifically under sample-size constraints. That is fine, but the investigators should have more replicates (i.e. more n=100 datasets) to provide a convincing case. Comparing these ML methods against real + simulated data would instill more confidence in their comparative study. The authors mentioned that they will seek to explore simulation in future work, but they did not do so for this manuscript. Small sample size with one dataset is still a problem here and the authors have not offered a solution in their response to the reviewers.

I am satisfied that the authors have done feature importance estimation as now mentioned in the results section and supplemental material. I am curious how this estimation was done for methods like ANN, where is it not always clear how features are used in hidden layers. It would be helpful if an extra sentence or two describing how feature importance estimation was done would be helpful. The authors now included Fig 1 in the supplemental material, but I don't have access to that as a reviewer so I cannot comment on that supplemental figure.

The reviewers did a good job at better describing their tunable parameters for all tested algorithms in the Methods section and Table 2. This increases the interpretability and reproducibility of this work.

The authors have made their R code publicly available and have provided a DOI link.

The lack of openly publishing these data may still be an issue. The authors cannot openly and freely share data for a variety of very good reasons, and they have provided alternative methods for investigators to request data on a case-by-case basis. Whether this meets the PLOS ONE standards and requirements for open data is up to the Editor and PLOS ONE.

7. PLOS authors have the option to publish the peer review history of their article (what does this mean?). If published, this will include your full peer review and any attached files.

Reviewer #1: No

Reviewer #3: **Yes: **Lucas Sheneman

---

## [Author Response · Author response to Decision Letter 1]

22 Sep 2022

We have revised the manuscript and adress the questions of the reviewers. 

No additional requirements were asked by reviewer 1.

Answer to reviewer 3: We appreciate the input and suggestions proposed by the reviewer. We agree with the reviewer that the small sample size could be the largest problem of the study. We also agree that one may think that these results could be biased for these small data sets and appreciate the suggestion of data augmentation creating different small data sets to replicate these results in the same conditions. Using the augmenteR R package we managed to create a oversampled augmented data where we performed resampling with replacement and built three small synthetic dataframes (n=100 each). In each of the synthetic dataframes we performed the same analysis and model comparisons. Similar results were found in the real data and synthetic dataframes with global better results in all the models in the synthetic dataframes (probably because the class imbalance were less). In conclusion, RF was the best fitting model in all the cases validating our previous results. This was added to methods, results, and discussion.

---

## [Decision Letter · Decision Letter 2]

29 Sep 2022

Machine learning outperformed logistic regression classification even with limit sample size: A model to predict pediatric HIV mortality and clinical progression to AIDS

PONE-D-21-31247R2

Dear Dr. Tagarro,

We’re pleased to inform you that your manuscript has been judged scientifically suitable for publication and will be formally accepted for publication once it meets all outstanding technical requirements.

Kind regards,

Zaher Mundher Yaseen

Academic Editor

PLOS ONE

Additional Editor Comments (optional):

Reviewers' comments:

Reviewer's Responses to Questions

**Comments to the Author**

1. If the authors have adequately addressed your comments raised in a previous round of review and you feel that this manuscript is now acceptable for publication, you may indicate that here to bypass the “Comments to the Author” section, enter your conflict of interest statement in the “Confidential to Editor” section, and submit your "Accept" recommendation.

Reviewer #1: All comments have been addressed

2. Is the manuscript technically sound, and do the data support the conclusions?

Reviewer #1: Yes

3. Has the statistical analysis been performed appropriately and rigorously? 

Reviewer #1: Yes

4. Have the authors made all data underlying the findings in their manuscript fully available?

Reviewer #1: Yes

5. Is the manuscript presented in an intelligible fashion and written in standard English?

Reviewer #1: Yes

6. Review Comments to the Author

Reviewer #1: We feel that the manuscript has extensively improved. ‎Thanks to the corresponding author, he was ‎successful in ‎answering our questions. (ACCEPT).‎

7. PLOS authors have the option to publish the peer review history of their article (what does this mean?). If published, this will include your full peer review and any attached files.

Reviewer #1: No

---

## [Editor Report · Acceptance letter]

5 Oct 2022

PONE-D-21-31247R2 

Machine learning outperformed logistic regression classification even with limit sample size: A model to predict pediatric HIV mortality and clinical progression to AIDS 

Dear Dr. Tagarro:

I'm pleased to inform you that your manuscript has been deemed suitable for publication in PLOS ONE. Congratulations! Your manuscript is now with our production department. 

Kind regards, 

on behalf of

Dr. Zaher Mundher Yaseen 

Academic Editor

PLOS ONE